# Hyper-Parameter Optimization of Stacked Asymmetric Auto-Encoders for Automatic Personality Traits Perception

**DOI:** 10.3390/s22166206

**Published:** 2022-08-18

**Authors:** Effat Jalaeian Zaferani, Mohammad Teshnehlab, Amirreza Khodadadian, Clemens Heitzinger, Mansour Vali, Nima Noii, Thomas Wick

**Affiliations:** 1Electrical & Computer Engineering Faculty, K. N. Toosi University of Technology, Tehran 19967-15433, Iran; 2Institute of Applied Mathematics, Leibniz University of Hannover, 30167 Hannover, Germany; 3Institute of Analysis and Scientific Computing, TU Wien, 1040 Vienna, Austria; 4Center for Artificial Intelligence and Machine Learning (CAIML), TU Wien, 1040 Vienna, Austria; 5Institute of Continuum Mechanics, Leibniz University of Hannover, 30823 Garbsen, Germany

**Keywords:** big five personality traits, cultural algorithm, deep learning, hyper-parameter optimization, personality perception

## Abstract

In this work, a method for automatic hyper-parameter tuning of the stacked asymmetric auto-encoder is proposed. In previous work, the deep learning ability to extract personality perception from speech was shown, but hyper-parameter tuning was attained by trial-and-error, which is time-consuming and requires machine learning knowledge. Therefore, obtaining hyper-parameter values is challenging and places limits on deep learning usage. To address this challenge, researchers have applied optimization methods. Although there were successes, the search space is very large due to the large number of deep learning hyper-parameters, which increases the probability of getting stuck in local optima. Researchers have also focused on improving global optimization methods. In this regard, we suggest a novel global optimization method based on the cultural algorithm, multi-island and the concept of parallelism to search this large space smartly. At first, we evaluated our method on three well-known optimization benchmarks and compared the results with recently published papers. Results indicate that the convergence of the proposed method speeds up due to the ability to escape from local optima, and the precision of the results improves dramatically. Afterward, we applied our method to optimize five hyper-parameters of an asymmetric auto-encoder for automatic personality perception. Since inappropriate hyper-parameters lead the network to over-fitting and under-fitting, we used a novel cost function to prevent over-fitting and under-fitting. As observed, the unweighted average recall (accuracy) was improved by 6.52% (9.54%) compared to our previous work and had remarkable outcomes compared to other published personality perception works.

## 1. Introduction

Whether deep or shallow, the operation of artificial neural networks (ANNs) depends on their hyper-parameters and parameters [1,2,3]. Certain variables of ANNs are called hyper-parameters, such as the number of layers [2], or control the training process, such as the learning rate [4]. In contrast, the trainable variables pertaining to layer connections and tuned during the training process, which are weights and biases, are called parameters [5,6,7]. Although parameter tuning may yield good results, it does not yield notable results without hyper-parameter tuning (HPT).

The importance of HPT became more manifest than before with the development of deep learning algorithms. Deep learning is a type of machine learning (ML) technique with diverse hyper-parameters that severely affect its performance [8,9,10]. Since HPT is an arduous task and requires data and network knowledge [11,12], it is often acquired by empirical methods (trial-and-error), which is time-consuming and does not guarantee significant results in terms of efficient algorithms and overall cost complexity. Therefore, studies based on applying optimization methods to ANNs have gained attention.

Accordingly, the usage of optimization algorithms is divided into three groups, as follows:HPT with the classical method and parameter optimization [13,14,15,16]: The fine-tuning of weights and biases (parameters) can provide useful information about the problem, but their size and initial value rely on HPT. Moreover, the number of parameters in deep neural networks (DNNs) and high dimensional datasets is enormous, and calculating the optimum value of these parameters is complicated, not easily implemented, and requires computational systems with remarkable capabilities.Hyper-parameter and parameter optimization [17,18,19]: Adaptive hyper-parameters are obtained by parameter training. The critical disadvantage is that with each possible vector of hyper-parameters, the parameters must be optimized, which causes runtime errors in the computational system and requires expensive training and large storage capacity to save the best parameters value over epochs. Additionally, all possible combinations of hyper-parameters are computationally infeasible. Hence, this method is not applicable in a large model such as deep learning [20,21].Hyper-parameter optimization (HPO) and parameter tuning with back-propagation [4,11,22]: The main drawback is that although optimization methods are efficient in finding global optima, the gradient may vanish when back-propagating. As a result, not all network parameters are tuned well, which impacts results [23]. To tackle the poor-tuning process of deep neural network parameters, an asymmetric auto-encoder (Asy_AE_) was presented in our previous work for automatic personality perception (APP) from speech [24]. We showed that Asy_AE_ could improve the model outcome results compared with conventional auto-encoders by semi-supervised training of parameters, and it can be effectively employed in deep learning. However, the stacked asymmetric auto-encoder (SA_AE_) hyper-parameters were chosen by trial-and-error, which was time-consuming, and two personality traits achieved lower accuracy than other prior research [24].

Thus, the aims of the present work were to (1) propose a novel optimization method based on cultural evolution and parallel computing, (2) obtain the near-optimal values of hyper-parameters of SA_AE_, and (3) classify five personality traits.

The rest of the article is organized as follows. In Section 2, some related works of HPO in deep learning and APP are explained. In Section 3, the dataset is introduced, and the summary of the feature extraction method is presented in Section 4. The new optimization method is proposed in Section 5. The simulation results of the new method, which is applied to three benchmark functions of finding global optima, are presented in Section 6. In addition, this section discusses the outcomes of applying the proposed method to SA_AE_ for automatic personality perception classification.

## 2. Related Works

Given that this article examines HPO methods in order to find a proper one to optimize the hyper-parameters of SA_AE_ for automatic personality trait perception, the related works section is divided into two parts. The focus of the first part is on recently published methods of neural network hyper-parameter tuning, regardless of the application in which it is used. Thus, the works related to the investigation of HPO in ML are summarized in the first part. Since the aim of our research was HPT of SA_AE_ to classify five personality traits from speech, the second part is related to studying HPO in machine learning methods applied in the field of personality trait perception.

### 2.1. Hyper-Parameter Tuning in ML

Deep learning hyper-parameter types are vast and can be divided into three groups: integer, real, and categorical. The integer group consists of variables such as the number of layers (whether hidden or convolutional) [25], the number of neurons [8], the size of the kernel [26], the number of kernels [27], batch size, pooling size, and number of maximum epochs [9]. The real group includes the learning rate [25], dropout rate [25], regularization factor [25], network weight initialization [5], and momentum [4]. The categorical group comprises activation function type [8] and optimization method [8].

Considering that a change in the value of each hyper-parameter changes the values of the neural network parameters that affect the output of the network, and also that examination of any possible combination of hyper-parameters is time-consuming, expensive and practically impossible, studies have investigated the effect of adjusting and optimizing some of the most important hyper-parameters.

In this regard, the article in [4] employed the HPO method for bearing fault diagnosis in mechanical equipment. Parallel computing was used to find hyper-parameters of the deep belief network (DBN). The learning rate and momentum were optimized, while other hyper-parameters were predefined and kept constant. Additionally, Wu Deng et al. used quantum-inspired differential evolution (DE) to optimize DBN parameters. Results showed an improvement in global search and avoiding premature convergence for fault classification [28].

The numbers of hidden neurons as a hyper-parameter and of the weights/biases as parameters were optimized in a feed-forward ANN by Gray wolf optimizer in [18]. Feed-forward ANNs (not back-propagation) were used because adjusted parameters were achieved by the optimization method.

Y. Peng et al. proposed an HPO method based on a fuzzy system in [8]. They optimized the number of hidden layers and the number of neurons in each layer of a DNN. The activation function type and optimization method, including Genetic Algorithms (GA), Bayesian search, grid search, random search, and quasi-random search, were selected automatically during HPO. For preventing over-fitting, the dropout technique was used. The proposed method was tested in three rainfall prediction datasets.

The authors of [29] suggested a distributed particle swarm optimization (PSO) for the HPO of a convolution neural network (CNN). They were concerned about the time-consuming population search based on distributed PSO, and parallel computing was employed to speed up the algorithm. They optimized the number and size of the kernels, the type of pooling (max or average) for two convolutional layers, the activation function type in convolutional layers, the number of neurons, learning rate, and the dropout rate of the fully connected layers.

Time-series prediction of congestion in highway systems based on long short-term memory (LSTM) was investigated in [9]. To obtain the proper model and structure, the authors recommended an HPO method by applying the Bayesian optimization (BO) method. Five hyper-parameters were automatically obtained, including learning rate, the number of hidden layers, the number of neurons in each layer, batch size, and dropout rate.

The intention of [25] was to examine the robustness of one HPO method over six benchmarks, contrary to other works that designed an algorithm that fit one problem. In other work, the authors used BO as an old HPO method in CNN [1] and applied four strategies to alleviate the drawbacks of BO. They tuned the hyper-parameters of two convolutional layers and two fully connected layers in this way.

In [26], an intuitive architecture design using GA was proposed for CNN. The obtained model was evaluated on a CNN with a single convolutional layer and a fully connected layer. Additionally, some hyper-parameters, including maximum epochs, batch size, initial learning rate, regularization, and momentum were optimized by PSO to prepare a CNN for expression recognition in [30].

Since the success of neural networks depends on their structure, the article in [31] proposed a micro-canonical optimization algorithm for overcoming large parameter spaces and optimizing hyper-parameters of a CNN. Hyper-parameters were the number of convolution layers, activation function type, batch size, pooling type, and dropout rate. The method was evaluated by six image recognition datasets and exhibited accuracy improvement.

State-of-health estimation and remaining usable life prediction in battery prognosis were examined in [32] by a deep convolution neural network. The authors addressed hyper-parameter tuning that affected DNN performance. They improved the algorithm by using the BO method.

Anjir A. Chowdhury et al. concentrated on the role of hyper-parameter optimization in the performance and reliability of deep learning outcomes [33]. They compared several HPO algorithms to obtain better validation accuracy in DNNs and concluded that most of them are computationally expensive. Finally, a greedy approach-based HPO algorithm was proposed for enabling faster computing on edge devices for on-the-fly learning applications. The VGG and ResNet architectures were used, and their hyper-parameters such as epochs, number of hidden layers, number of units per layer, activation function, dropout rate, batch size, and learning rate were optimized.

The Gray wolf optimization was employed to optimize the parameters of the kernel extreme learning machine to realize a hyperspectral image classification method in [34].

### 2.2. Automatic Personality Perception

In psychology, the big five inventory (BFI) is a well-known theory of personality with five traits, including openness to experience (Ope.), conscientiousness (Con.), extraversion (Ext.), agreeableness (Agr.), and neuroticism (Neu.). These traits are in an individual simultaneously by different scores and can be measured by a BFI questionnaire in general [35,36].

Due to the importance of personality in daily life, computer science researchers have investigated personality trait identification by multimodal media (audio, text, video, image) recently. Here, we focus on studies structured by deep learning methods.

A multimodal approach for perceiving personality traits was proposed by employing well-known deep structures (ResNet-v2-101 and VGGish) [37]. The LSTM network for using temporal information was added at the end. The authors optimized only the learning rate, while other hyper-parameters were configured manually. It is clear that the structure of the mentioned deep methods is fixed, and the weights and biases are pre-trained. Therefore, HPO or HPT does not tune according to each dataset in these networks.

Given the fact that personality traits can influence appearance, MobileNetv2 and ResNeSt50 networks were employed in [38] to extract facial features and classification. Results specified that one pre-trained network such as MobileNetv2 is inappropriate for classifying all five personality traits. It indicated that each trait must classify by a specific model, which means different hyper-parameters are necessary. However, the authors did not mention it directly and applied a combination of two pre-trained deep networks to build a complex deep model.

Onno Kampman et al. examined feature extraction and the classification of five personality traits by applying a one-dimensional CNN to a raw audio dataset. The HPT of the deep network containing regularization factors and kernel size was performed manually [39].

One of the personality detection applications is discovering interpersonal communication skills. Article [40] investigated this aspect from a video interview using a semisupervised CNN in which HPT was performed by trial-and-error. The authors concentrated on video processing, and a fixed hyper-parameter set to utilize for all traits.

The study in [41] analyzed the acoustic and lexical features of a speech signal that were affected by BFI traits. Additionally, it designed six models based on recurrent neural networks for classifying those traits. Hyper-parameters such as hidden size, learning rate, batch size, and dropout percentage were defined, but tuning them was not discussed.

## 3. Dataset

The SSPNet speaker personality corpus (SPC) is a well-known automatic personality perception dataset introduced in 2010. This dataset originally contained 640 recorded speech signals of 322 native French speakers. There is one speaker in each clip recorded for 10 s. Due to the studies on the effect of mental factors on speech signals [42], the collected clips were emotionally neutral, and to confirm that lexical content did not affect the personality scores, evaluators who were foreign to the French language were selected. Therefore, eleven assessors who did not understand French evaluated each clip based on the BFI questionnaire. The average score of these assessors was considered as the final score for each clip. Hence, five scores were obtained for each clip [43].

Although the SPC dataset has been applied in several works and is a proper dataset for comparison with the new methods, the number of samples is uses is low to train the enormous number of parameters of a DNN. This important challenge was addressed in our previous work [24], and we proved that the sample size of speech signals could be enhanced with data augmentation methods based on a spectrogram so that the prosodic content of speech could be preserved. Data augmentation is a popular technique to expand the size of the dataset artificially and is widely used in image processing. However, using this technique in speech is not as easy as using an image. In other words, we needed to choose transformations that maintain the speaker’s personality, and we had to be confident that such manipulations in the spectrogram do not interfere with the extracted features related to personality traits. In this regard, frequency masking and time warping were selected as data augmentation methods, and the number of clips increased up to 640,000. For more details, please see [24].

## 4. Feature Extraction

Despite DNN’s ability to perform automatic feature extraction from raw speech signals, deep learning methods have been generally applied to manually extracted hand-crafted audio features. This is mainly because of the large volume of data required for deep learning methods to outperform. Nevertheless, building a dataset with large available labeled samples is costly, time-consuming, and laborious work in the automatic personality perception field, which restricts various methods. Therefore, previous studies have used handcrafted features for the DNN input [44].

These handcrafted features contain 6373 statistical features extracted from 130 low-level descriptions (LLD) [45]. Table 1 contains 65 LLD features and 65 first derivatives of LLD (ΔLLD), for a total of 130 LLD features.

For the LLD feature extraction process, each clip was divided into 60 ms frames with a 20 ms overlap in the time domain and 20 ms frames with a 10 ms overlap in the frequency domain by the Opensmile2.3 toolkit.

## 5. Proposed Method

This section is divided into two parts. In the first part, we thoroughly describe the new optimization method mathematically. In order to apply our optimization method to the SA_AE_, we had to address several problems. The second part deals with this issue and its solution.

### 5.1. The Proposed Optimization Method

HPO of deep learning is a time-consuming task in practice that depends on the network depth, the size of parameters, processor system, and optimization algorithm speed [5]. Applying HPO to deep learning is challenging. It can be (1) the unsupervised learning of most deep learning methods that causes trouble for optimization and imperfect tuning of parameters [47], (2) a large model with enormous trainable parameters that lead the processing system to runtime errors [5,8], and (3) an intricate search space created by different types of hyper-parameter domains (categorical, continuous, and integer value), causing inherent computational complexity [5]. A larger search space gives rise to a longer search time.

Parallel evaluation can partly reduce optimization time [48], and culture speeds up the population’s evolution more than chromosomes (each chromosome represents a solution in the population space) [49]. Accumulated experience that is potentially accessible to all individuals is called culture, which is used in problem-solving activities [50]. The knowledge extracted by identifying patterns in the population’s problem-solving experiences influences the generation of new solutions [51]. Therefore, the combination of CA and parallel computing can facilitate the discovery of the search space [52]. In this regard, researchers are interested in combining CA with other optimization algorithms. Sun et al. combined a cultural algorithm and two PSO populations and shared their belief space. It indicated that sharing knowledge of belief space can improve performance by avoiding local optima [53]. A single population and multi-population based on CA was proposed in [54]. A PSO population-based method with interactive belief space was introduced by [49]. A hybrid evolutionary optimization method coupling CA with GAs was defined in [55]. Fuzzy operations were employed to exchange individuals between belief space and population space in [56].

From this perspective, we proposed a four-island approach based on the parallel evaluation and CA.

Although CA and parallel computing can perform better than the basic optimization algorithms [57], they do not provide enough convergence speed alone for deep learning. Thus, three driving force factors were applied to population space for creating interactive space between four island population spaces. Creating interactive population space causes interactive belief space, which can determine the direction and step size faster than traditional optimization methods. In this regard, our proposed method is called the multi-island interactive cultural (MIC) algorithm.

The MIC method is illustrated in Figure 1. In this method, control parameters are configured firstly. The initial population X[m, D] is generated randomly in the feasible space. The variable m indicates the population size (the number of chromosomes or individuals), and D is chromosome dimension (the number of genes).

After preparing the random initial population, it transfers into the four islands in parallel (gray lines): GA, PSO, DE, and evaluation strategy (ES). The GA and PSO are the optimization algorithms widely applied to HPO studies in deep learning [1,8]. GA is far more successful in complex networks such as CNNs, but eliminates previous information by changing the population every iteration [50]. PSO shares information between the particles and is popular on the smaller networks [29]. The DE algorithm is utilized in optimization problems due to the high convergence speed and low control parameters when searching global optima. It is suitable for nonlinear search spaces [28]. The ES is less popular among the global optimization algorithms because it is a simple mutation-selection method, but it is helpful in making small changes [48]. It should be noticed that in the first iteration, the population of the four islands is the same.

The four islands were evaluated individually and in parallel. Then, some individuals of each island were randomly selected to transfer into an interactive belief space (InBS) through an acceptance function (colored arrows). Here, the acceptance function was 25% of the best individuals of each island. So, the belief space size was y[m, D].

The InBS consisted of normative ($N\left[D\right]$) and situational knowledge (S) of all islands. Knowledge of different islands in the belief space causes the chromosomes to move away from unwanted regions and get closer to the optimal points by using different experiences faster than previously published works. InBS can be used effectively to prune the population space.

Normative knowledge represents the range of the best solutions by determining the upper and lower bands of each gene of a chromosome and is used to influence the direction of the search efforts within the promising ranges. In other words, it computes the range of each gene that leads the individual to a good solution.

The offspring affected by normative knowledge are generated by Equation (1) as
(1)$$y_{p+i,j}^{t+1}=\left\{\begin{array}{ll} y_{i,j}^{t}+|(u_{{}_{j}}^{t}-l_{{}_{j}}^{t})*N(0,1)| & if\;\;\;y_{i,j}^{t}<l_{{}_{j}}^{t},\\ y_{i,j}^{t}-|(u_{{}_{j}}^{t}-l_{{}_{j}}^{t})*N(0,1)| & if\;\;\;y_{i,j}^{t}>u_{{}_{j}}^{t},\\ y_{i,j}^{t}+\beta |(u_{{}_{j}}^{t}-l_{{}_{j}}^{t})|*N(-1,1) & \mathrm{otherwise}, \end{array}\right.$$
where $u_{j}$ is the upper and $l_{j}$ is the lower band of InBS for jth gene, respectively, *β* is a constant value, t is the current iteration, and N(0,1) is the normal distribution.

For each gene, the structure contains the upper band ($u_{j}^{t}$), the lower bound ($l_{j}^{t}$), the upper band value ($U_{j}^{t}$), and the lower bound value ($L_{j}^{t}$), which are obtained by Equations (2)–(5), respectively.
(2)$$L_{j}^{t+1}=\left\{\begin{array}{ll} f(y_{i}) & if\;\;\;y_{i,j}\leq l_{j}^{t}\;\;\;Or\;\;\;f(y_{i})<L_{j}^{t},\\ L_{j}^{t} & \mathrm{otherwise}, \end{array}\right.$$
(3)$$l_{j}^{t+1}=\left\{\begin{array}{ll} y_{i,j}^{t} & if\;\;\;y_{i,j}^{t}\leq l_{j}^{t}\;\;\;Or\;\;\;f(y_{i}^{t})<L_{j}^{t},\\ l_{j}^{t} & \mathrm{otherwise}, \end{array}\right.$$
(4)$$U_{j}^{t+1}=\left\{\begin{array}{ll} f(y_{i}) & if\;\;\;y_{i,j}\geq u_{j}^{t}\;\;\;Or\;\;\;f(y_{i})<U_{j}^{t},\\ U_{j}^{t} & \mathrm{Otherwise}, \end{array}\right.$$
(5)$$u_{j}^{t+1}=\left\{\begin{array}{ll} y_{i,j}^{t} & if\;\;\;y_{i,j}^{t}\geq u_{j}^{t}\;\;\;Or\;\;\;f(y_{i}^{t})<U_{j}^{t},\\ u_{j}^{t} & \mathrm{otherwise}, \end{array}\right.$$
where $y_{i,j}$ is the jth gene in the ith individual of InBS, and the $f(y_{i})$ is the value of the individual $y_{i}$ calculated by the fitness function. A fitness function (loss function) evaluated individuals of each island separately. The problem description determines the fitness function.

The situational knowledge, as seen in Equation (6), adjusts the mutation step size relative to the distance between the current best individual and the other individuals. The greater the distance between ith individual, y_i_, and the current best individual, the greater the step size and vice versa.

Updating the situational knowledge adds the InBS’s best individual to the situational knowledge if it outperforms the current best individual, as described in Equation (6).

Here, $y_{best}^{t}$ is the best individual in the InBS at iteration t. The influence rule can be represented with Equation (7) (for i = 1, …, m and j = 1, …, D).
(6)$$<E_{1}^{t+1},E_{2}^{t+1},\ldots ,E_{e}^{t+1}>=\left\{\begin{array}{ll} <y_{best}^{t},E_{2}^{t},\ldots ,E_{e}^{t}> & if\;\;\;f(y_{best}^{t})>f(E_{1}^{t}),\\ <y_{best}^{t}> & if\;\;\;\mathrm{change}\;\;\mathrm{detected},\\ <E_{1}^{t},E_{2}^{t}\ldots ,E_{e}^{t}> & \mathrm{otherwise}, \end{array}\right.$$
(7)$$y_{{}_{p+i,j}}^{t+1}=\left\{\begin{array}{ll} y_{{}_{i,j}}^{t}+|(y_{{}_{i,j}}^{t}-E_{{}_{i,j}}^{t})\cdot N_{i,j}(0,1)| & if\;\;\;y_{{}_{i,j}}^{t}<E_{{}_{j}}^{t},\\ y_{{}_{i,j}}^{t}-|(y_{{}_{i,j}}^{t}-E_{{}_{i,j}}^{t})\cdot N_{i,j}(0,1)| & if\;\;\;y_{{}_{i,j}}^{t}>E_{{}_{j}}^{t},\\ y_{{}_{i,j}}^{t}+\beta |(y_{{}_{i,j}}^{t}-E_{{}_{i,j}}^{t})|\cdot N_{i,j}(0,1) & \mathrm{otherwise}, \end{array}\right.$$
where $E_{j}$ is the jth gene in the best individual, β is a constant factor, N(0, 1) is the normal distribution, and $y_{p+i,j}$ is the offspring of the individual $y_{i,j}$.

After updating InBS with new generations, some individuals are transferred into each island population space by influence function. There is no doubt that the individuals of InBS contain the knowledge of all of the islands. This is the ability of the proposed method. Various studies have shown that the efficiency of optimization methods is altered for different problems. In other words, choosing an optimization method for a problem is a challenge that some researchers consider as a kind of hyper-parameter that needs to be tuned. Hence, 25% of the best individuals of InBS were replaced with 25% of the worst population on each island. Offspring generation processing is started in each island separately and evaluated through fitness function.

If the algorithm reaches the stopping criterion, the process will be stopped. Otherwise, interactive population space is created by three driving forces in order to promote cooperation among the islands and increase diversity.

The three driving-force methods are named the elitism method (EM), merge method (MM), and lambda method (LM).

In interactive population space, all individuals of each island are considered. In EM, the best individuals with size m are preserved and replaced with the old population on each island. As we use this method, the populations of the next generation for each island are the same. This driving force method forces the four basic algorithms to create interactive space only by the best individuals of four islands.

In MM, after considering all individuals of each island, a random number *a*, *a* ∈ (0, 1), is produced. The *a* × *m* $\left(a*\mathrm{sizeofpopulation}\right)$ of the best individuals are merged with (*a* − 1) × *m* of the old population on each island. It is clear that each island has a unique new population in this interactive space.

In LM, two of the islands are selected randomly, according to two random numbers µ, µ ∈ (0, 1), and λ, λ ∈ (0, 1), representing emigration and immigration, respectively. The random numbers of individuals based on µ and λ of each island indicate which individuals can immigrate to and emigrate from another random island. This method forces islands to cooperate with the best individual and the worst one to create interactive space.

Due to the interaction and sharing of individuals among the four islands, if one algorithm traps in local optima, others can lead MIC into global optima because the result is not dependent on a single algorithm. This feature allows the MIC to be used for various global optimization problems to escape local optima efficiently.

The MIC strategy is presented step by step below (Algorithm 1).
**Algorithm 1:** Implementation of MIC**Step 1**: Set the MIC parameters randomly.**Step 2**: Generate the initial population randomly.**Step 3**: Transfer 25% of the best individuals of each island into InBS (Accept). **Step 4**: Update Belief space whith Equations (1)–(7).**Step 5**: Transfer 25% of offspring into each island (Influ).**Step 6**: **If** stop criterion < ζ      Stop algoriyhm.**    Else**      Go to Step 7. **Step 7:** Create Interactive population space by using the following three methods:     EM: *m* of the best individuals of four islands are selected and replaced with an old population.      MM: The *a × m* $\left(a*sizeofpopulation\right)$ of the best individuals are selected and merged with (*a* − 1) × *m*, which is obtained from the old population in islands.     LM: According to two random numbers, µ and λ, some individuals of a random island can immigrate to and emigrate from another random island.**Step 8:** Go to Step 3.

### 5.2. Stacked Asymmetric Auto-Encoder HPO Using MIC

Since our work aimed to obtain the SA_AE_ near-optimal structure, a brief overview of this method is presented below.

(1)
**Stacked asymmetric auto-encoder**


The Asy_AE_ is a semi-supervised DNN that poses the curse of dimensionality. The schematic of the Asy_AE_ is illustrated in Figure 2.

In this type, one neuron is added in the decoder part of the conventional auto-encoder with the desired value of the problem, which is the studied personality score in our field. The symmetry of the encoder and decoder parts is disrupted by this single neuron and made asymmetric.

The feed-forward equations of the Asy_AE_ are similar to the conventional one as follows. For representing encoder and decoder layers, superscripts of 1 and 2 were used, respectively.
(8)$$net^{\left(1\right)}=W^{\left(1\right)}X,$$
(9)$$O^{\left(1\right)}=f\left(net^{\left(1\right)}\right),$$
where $W^{\left(1\right)}$ indicates the encoder weight matrix, X displays the input matrix, **O**^(1)^ is the encoder output matrix, and f is the activation function.
(10)$$net^{\left(2\right)}=W^{\left(2\right)}O^{\left(1\right)},$$
(11)$$O^{\left(2\right)}=f\left(net^{\left(2\right)}\right),$$
where $W^{\left(2\right)}$ and **O**^(2)^ are the weight and output matrixes of the decoder layer, respectively.

The error back-propagation related to the encoder and decoder weights matrixes is calculated by Equation (12).
(12)$$E:=\frac{1}{k}\sum_{i=1}^{k}\log\left(\cosh\left(e_{t}\right)\right),$$
where $e_{t}$ is the error vector of Asy_AE_ at time t, which is described by Equation (13), and k is the neuron size of decoder layer output.
(13)$$e_{t}:=d_{t}-\;o_{t}{}^{\left(2\right)}.$$

The desired output vector at time t is presented by $d_{t}$, which belongs to the matrix D. It is the desired output matrix of Asy_AE_, which is produced by the combination of desired labels and Asy_AE_ input.
$$D=\left[\begin{array}{cccc} x_{11} & x_{12} & \ldots & x_{{1n}_{0}}\;\;\;\;L\\ x_{21} & x_{22} & \ldots & x_{{2n}_{0}}\;\;\;\;L\\ \vdots & \vdots & \vdots & \vdots \\ x_{m1} & x_{m2} & \cdots & x_{{\mathrm{mn}}_{0}}\;\;\;\;L \end{array}\right].$$

Here, x_ij_ is the Asy_AE_ input matrix element, and L is the desired label of the problem. A stacked asymmetric auto-encoder is a result of putting several Asy_AE_s together.

(2)
**Optimizing some hyper-parameters of a stacked asymmetric auto-encoder**


Given the fact that the number of DNN hyper-parameters is significantly large, the simultaneous optimization of all of them complicates the computation and requires high-performance computing systems. Hence, we compromised between MIC and expertise for calculating the six critical DNN hyper-parameters as follows:number of neurons in each hidden layerlearning rate valueinitial parametersnumber of hidden layersmaximum epoch of network trainingpreventing over-fitting and under-fitting

For HPO of SA_AE_, the following principles come after. Figure 3 illustrates the flowchart of the proposed method in detail.

**Determining the number of neurons in each hidden layer:** In our work, $N_{i}$ indicates the number of neurons in the $i^{\mathrm{th}}$ hidden layer that will be optimized by the MIC method. So, the first variable of MIC is $N_{i}$, which is an integer value, $N_{i}\in \left[1,m\right]$ where m value is equal to the input size of Asy_AE_. It forces the Asy_AE_ to be an incomplete network. It means the encoder layer has fewer neurons than the input layer.

**Determining the learning rate in each hidden layer:**$\mu_{i}$ specifies the learning rate in the $i^{\mathrm{th}}$ hidden layer, which will be optimized by the MIC method. Therefore, the second variable of the MIC population is a real value between zero and one, ${\;\mu }_{i}\;\in \left(0,1\right)$. It should be mentioned that we set the decimal digit of ${\;\mu }_{i}$ equal to 5 to examine its effect on SA_AE_ performance.

**Initial value of trainable parameters:** Although deep learning methods have good performance in various problems, they are complicated tasks. Because there are huge factors that strongly influence them, one of the critical factors is initialization.

The DNN parameters need a starting point in the feasible area to be trained. The proper initial parameters can accelerate the convergence. Contrarily, random initialization can trap the network in the local optima.

Optimization algorithms such as GA and PSO can be used in this field. However, the number of DNN parameters (weights and biases) is vast, e.g., 10^15^, and producing the chromosomes with these dimensions causes a memory error in the processor system and is not efficient in practice. Another method, suggested by Hinton et al., applies the restricted Boltzman machine (RBM) network to tune the auto-encoder’s initial parameters [58,59].

According to the ANN-base of an Asy_AE_ and RBM, the Asy_AE_ can be interpreted as two consecutive RBMs illustrated in Figure 4. The input layer is the visible unit, and the encoder layer is the hidden unit for the RBM_1_. In the RBM_2_, the encoder layer is the visible unit, and the decoder layer is the hidden unit.

The conventional RBM is based on binary visible and hidden units, called Bernoulli-Bernoulli RBM (BBRBM). If both visible and hidden units have a Gaussian distribution, the Gaussian-Gaussian RBM (GGRBM) is employed [60]. Since the Asy_AE_ input and parameters are real values, we used the GGRBM equations.

The energy function of the GGRBM is defined as Equation (14), where v presents visible units and h shows hidden units. It should be noted that the Asy_AE_ input and the encoder output are the visible units of RBM_1_ and RBM_2_, respectively.
(14)$$E(v,h)=-\sum_{i=1}^{g_{v}}\sum_{j=1}^{g_{h}}W_{i,j}\frac{v_{i}h_{j}}{\sigma_{i}\sigma_{j}}-\sum_{i=1}^{g_{v}}\frac{{\left(v_{i}-a_{i}\right)}^{2}}{2\sigma_{i}^{2}}-\sum_{j=1}^{g_{h}}\frac{{\left(h_{j}-b_{j}\right)}^{2}}{2\sigma_{{}_{j}}^{2}},$$
where $a_{i}$ and $b_{j}$ are visible and hidden units biases, respectively, $\sigma_{i}$ and $\sigma_{j}$ are their standard deviations. $W_{i,j}$ is the weight between the visible and hidden units. A probability value is assigned to each possible visible and hidden unit by Equation (15),
(15)$$P(v,h)=\frac{1}{Z}\exp(-E(v,h)).$$

Here, Z is the normalization constant calculated by Equation (16).
(16)$$Z=\sum_{v}\sum_{h}\exp(-E(v,h)).$$

Equation (17) shows the loss function, which must be maximized,
(17)$${\mathrm{maximize}}_{\left\{W_{i,j},a_{i},b_{j}\right\}}\frac{1}{c}\sum_{L=1}^{c}\log\left(P\left(v^{L},h^{L}\right)\right),$$

The updating functions are
(18)$$\Delta W_{i,j}=\zeta \left(<v_{i}h_{j}{>}_{data}-<v_{i}h_{j}{>}_{\mathrm{model}}\right),$$
(19)$$\Delta a_{i}=\zeta \left(<v_{i}{>}_{data}-<v_{i}{>}_{\mathrm{model}}\right),$$
(20)$$\Delta b_{j}=\zeta \left(<h_{j}{>}_{data}-<h_{j}{>}_{\mathrm{model}}\right),$$
where $<•{>}_{data}$ and $<•{>}_{\mathrm{model}}$ are expanded values of sample data and model probabilistic distribution, and ζ is the learning rate.

We described GGRBM briefly, and this is the time to use it. For a traditional auto-encoder, first, the initial parameters of the encoder layer are randomly selected and then trained by the GGRBM method. The trained parameters are considered the encoder layer’s initial parameters, and its transposition is employed for the decoder layer. However, in the Asy_AE_, the encoder and decoder parameters are not symmetric and have to be obtained individually. So, the above principle is applied to the decoder layer to obtain the initial parameters.

**The number of hidden layers****:** The value of this hyper-parameter is dependent on the performance of Asy_AE_s. The classification performance of each Asy_AE_ is examined in MIC for each pair of ($N_{i}$,${\;\mu }_{i}$). For the next Asy_AE_, the performance has to be better than that of the previous one. If the performance of Asy_AE(i+1)_ is better than that of Asy_AE(i)_, the MIC algorithm is continued.

The performance criterion is different from one problem to another. The Unweighted Average (UA) recall criterion frequently used in personality perception studies is calculated by Equation (21),
(21)$$\mathrm{UA}\;\mathrm{recall}=\frac{1}{2}\left({\mathrm{recall}}_{\mathrm{Low}}+{\mathrm{recall}}_{\mathrm{High}}\right),$$

The ${\mathrm{recall}}_{\mathrm{Low}}$ means the recall of detecting the low degree of studied personality, and the ${\mathrm{recall}}_{\mathrm{High}}$ indicates the recall of detecting its high degree.

**The maximum epoch of network training:** Generally, the DNN training process proceeds to reach maximum epoch (updating time) [40]. As discussed in [24], proper data separation does not occur in the maximum epoch. Thus, a J variation is employed as a stopping criterion to finish the training process in the epoch in which the maximum separation is achieved.

J is calculated as follows,
(22)$$J=\frac{\det(S_{B})}{\det(S_{W})},$$
where S_W_ is a within-class scattering matrix, and S_B_ is a between-class scattering matrix [61]. **det** represents the determinant of a matrix.
(23)$$S_{w}=\sum_{i=1}^{c}\sum_{x\in c_{i}}(X-\mu_{i}){\left(X-\mu_{i}\right)}^{T},$$
(24)$$S_{B}=\sum_{i=1}^{c}n_{i}(\mu_{i}-\mu ){\left(\mu_{i}-\mu \right)}^{T}.$$

Here, $n_{i}$ is the instance number of ith class, X is the encoder output matrix, and c is the number of classes, µ is the matrix for average all instances, and μ_i_ is the class average matrix of ith class.

**Preventing over-fitting and under-fitting problems:** The over-fitting problem happens when a model trains properly on the training dataset but performs poorly on the testing dataset. The under-fitting problem occurs when a model performs poorly on both the training and testing samples.

The number of layers and the neurons in each layer can excessively lead a model into over-fitting or under-fitting. This can be easily changed by changing the structure. More neurons and layers complicate the model, but fewer cannot pursue the data pattern. Therefore, this is one of the problems that has to be dealt with in designing an optimum structure. So, a new loss function is defined to guide the model toward good fitting.
(25)$$\mathrm{Loss}=\frac{{\mathrm{UA}}_{\mathrm{train}}}{a}*\frac{{\mathrm{UA}}_{\mathrm{val}}}{b},$$
where a is the training threshold, and b is the validation threshold. We already discussed the UA recall criterion used chiefly in personality perception. We applied the loss function defined in Equation (25) instead of Equation (21). The aim is the maximization of Equation (25). We set a = 0.8 and b = 0.6 because a UA_train_ of more than 80% and UA_val_ of more than 60% are acceptable. The loss value can be in the range of [2.08, 0]. So, the set of ($N_{i}$,${\;\mu }_{i}$) is acceptable to be maximized in Equation (25).

**Final algorithm:** The pseudo-code of optimizing SA_AE_ hyper-parameters is described in Algorithm 2.
**Algorithm 2:** Optimizing SAAE hyperparameters **Set** the initial parameters Old_max = 0, G_max = 2.08 (upper band of Loss), OldEv_Asy = 0 (the first Asy_AE_ performance) and the other randomly.**Set** the input matrix of Asy_AE_.**Set**i = 1 (i indicates the number of hidden layer)**Set** NewEv_Asy = 1 (the $\left(i+1\right)$th Asy_AE_ performance)**While** NewEv_Asy> OldEv_Asy    OldEv_Asy = NewEv_Asy  ** While** (G_max-Old_max) > 0.1      Optimize ($N_{i}$,${\;\mu }_{i}$) with MIC.      Initialize the Asy_AE_ parameter randomly.      Tune Asy_AE_ initial parameters with GGRBM.      Train Asy_AE_ while J increases.      Evaluate Equation (25).      **If** the value of Equation (25) ≥ Old_max       Old_max = the current value of Equation (25),       NewEv_Asy = the value of Equation (25).      **End if**
      Set the encoder layer output of ith Asy_AE_ as the input of $\left(i+1\right)$th Asy_AE_.      i = i + 1.  ** End while**
**End while**

## 6. Simulations and Results

In this section, firstly, the results of the MIC method on three benchmarks and comparison with other published methods will be discussed. Then, the MIC will be used to design the structure of five individual DNNs for classifying five personality traits. A final comparison can be found at the end of this section.

### 6.1. The Results of the MIC on Three Optimization Benchmarks

Three well-known, multimodal, continuous, and non-separable benchmark functions that have a global minimum value of zero, called Rastrigin [52], Ackley [62], and Griewang [62], are used to validate the MIC method.

The multimodal property means having many local optima or peaks in the function, which can test the ability of an algorithm to avoid being stuck in a local minimum. Non-separable refers to the independence of obtained solution variables. If all variables are independent, they can be optimized independently, and the function will be optimized [62]. Therefore, these three functions are complex problems in evaluating the performance of any new optimization algorithm.

The formula, feasible range of variables, and the global optima points of three functions are summarized in Table 2.

Here, n indicates the dimension of the function, which is n≥2 for all mentioned functions.

Figure 5 shows the shape of the functions described in Table 2. As can be seen, all three functions have many local optima and are suitable to show the ability of optimization methods to escape from being stuck in local optima.

In order to show the performance of MIC against the conventional optimization methods, the comparison results of the mentioned four islands and MIC are reported in Table 3.

Given the fact that the problem complexity increases with increasing dimensionality, increasing the number of the variables (dimension) grows the search space, which makes exploring the best solution difficult [62]. To investigate the effect of dimension on searching quality in MIC, we compared our results with 30D and 10D in Table 3.

For a fair comparison, all parameters and initial populations for the basic algorithms and MIC were set to the same values.

The following six criteria were utilized for a more reliable analysis. It should be mentioned that these criteria are common in optimization problems.

The average of iterations where the stop criterion is reached for examining convergence speed (AvI).The average of obtained best optima point (AvP).The smallest iteration at which the stop criterion occurs (SI).The best-obtained optima point (BOP).Calculating the standard deviation (SD) for proving the efficiency and robustness of the algorithm.The number of successful runs divided by the total number of runs called success rate (SR).

Table 3 shows the simulation outcomes of MIC and four basic optimization algorithms. 

It was concluded by AvI numerical results that MIC can reach more accurate solutions with a faster convergence speed than traditional algorithms in n = 10. Although for the n = 30, the MM performance diminishes, LM and EM preserve their performance with increasing complexity. It is demonstrated that LM and EM improve solutions steadily for a long time without getting stuck in local minima. It is clear that MIC is more powerful than the four basic algorithms alone when it comes to solving global optimization problems.

According to the AvP values in n = 10 and n = 30, traditional algorithms are often unsuccessful in finding favorable solutions in comparison to MIC, especially EM. Additionally, it can be concluded from AvP that the MIC speeds up the convergence to the global optima. The AvP values in n = 30 in comparison to n = 10 decreased about 0.1 in Rastrigin and remained constant for the other two functions in LM and EM. The change in the AvP values in MM is meaningful, which indicates getting stuck in the local optimum with the increase in the complexity of the problem, like the traditional methods.

Our SI outcomes show that the MIC method, especially EM and LM, reaches the stop criterion in a few iterations. It means the MIC method speeds up convergence. Moreover, the SI criterion shows that although the MM method performs better than the basic optimization methods in simpler functions (n = 10), its performance drops in complex functions (n = 30). LM and EM not only show their effectiveness in simple functions, but also perform well in complex problems compared to other methods.

The evaluation results of criterion BOP show that the LM and EM methods achieve the global optimal value more accurately than the basic methods in n = 10 and n = 30. However, MM implementation results decrease with increasing complexity.

It can be seen that the SD values of MIC, except for MM, are very small in comparison to those of the four basic algorithms in n = 10 and n = 30, which means the repeatability and robustness of the new algorithm are due to pruning search space.

The SR results prove that the MIC is very promising in bringing higher reliability than traditional algorithms because the number of times that LM, EM, and MM reached the desired value of the function was 100% in n = 10. As can be seen, as the complexity of the function increases (n = 30), the LM and EM methods are still successful in reaching the desired value.

From Table 3, it is concluded that despite increasing dimensions, the implementation outcomes of all algorithms decrease, except EM and LM.

Our study indicates that the quality of the solutions found using our proposed method for widespread global optima functions is higher than that of the solutions provided by traditional algorithms. This is due to a more appropriate tradeoff between exploring new individuals and exploiting highly fit individuals found at the parallelism level. By means of three widespread test functions, it is demonstrated that the new method has great potential for substantial improvement in search performance.

Due to the wide usage of these benchmarks, a comparison with other published works is presented in Table 4. It can be observed that LM and EM achieved the best solution in Ackley and Griewang functions (30D).

### 6.2. The Results of Personality Perception with The MIC Method

After the successful outcomes with the MIC method to find the global optima of three complex benchmark functions, we applied our novel method to find the near-optimal values of hyper-parameters for classifying five personality traits. We used “near-optimal” instead of “optimal” structure because tuning of MIC hyper-parameters such as mutation and crossover rating is chosen randomly.

Taking into account that different personality traits have different effects on speech characteristics [24,42], using the same DNN structure for all traits to extract features is not recommended. Assuming the five personality traits were independent, five separate neural networks were designed and trained to classify the five personality traits.

Hence, the network’s depth was determined by classifying the output of each Asy_AE_ encoder layer by the SVM with radial basis function kernel. The Asy_AE_ with higher classification results is considered as the output layer of the S_AE_.

Table 5 shows the comparison results of our proposed method with other works in the SPC dataset in terms of UA recall and accuracy. In our previous work, the structure of SA_AE_ was chosen by trial-and-error, which was time-consuming, and two traits (extraversion and openness) achieved lower accuracy than reported by other research [24]. N/A means not available.

In the present study, not only were the accuracy of extraversion and openness improved, but UA recalls were also increased more than before. This evidences that the performance and robustness of trained models are highly dependent on their hyper-parameter settings.

## 7. Conclusions

Since HPT is the most challenging aspect of ANN studies, it is mostly obtained by trial-and-error, affecting its performance. This article proposed a new approach based on cultural evolution and parallel computing to achieve a near-optimal structure of SA_AE_ in a reasonable time for automatic personality perception. We used the concept of parallelism and information on different regions of the search space to improve the search spaces in MIC and exchanged them between islands to provide greater population diversity. The proposed approach was implemented on three complex benchmarks, and six criteria evaluated our method’s performance in comparison with four basic optimization methods. The results showed that our approach outperforms other traditional optimization and newly published algorithms in four aspects: (1) convergence speed, (2) precision, (3) escaping from entrapment in local optima, and (4) repeatability. As an indication of our method’s performance, we increased the problem complexity by increasing the number of variables up to 30. The outcomes demonstrated the reliability of the MIC method, especially LM and EM. Subsequently, five hyper-parameters of SA_AE_ were optimized. Since the tuning of hyper-parameters affects over-fitting and under-fitting, we introduced a new cost function to control them during the optimization process.

In comparison with the results of our previous published work, the outcomes of applying MIC to SA_AE_ indicated 3.3% (3.1%) for consciousness, 5.1% (7.5%) for agreeableness, 5.9% (14.3%) for openness, 5.6% (10.1%) for extraversion, and 12.7% (3.6%) for neuroticism.

## Figures and Tables

**Figure 1 sensors-22-06206-f001:**
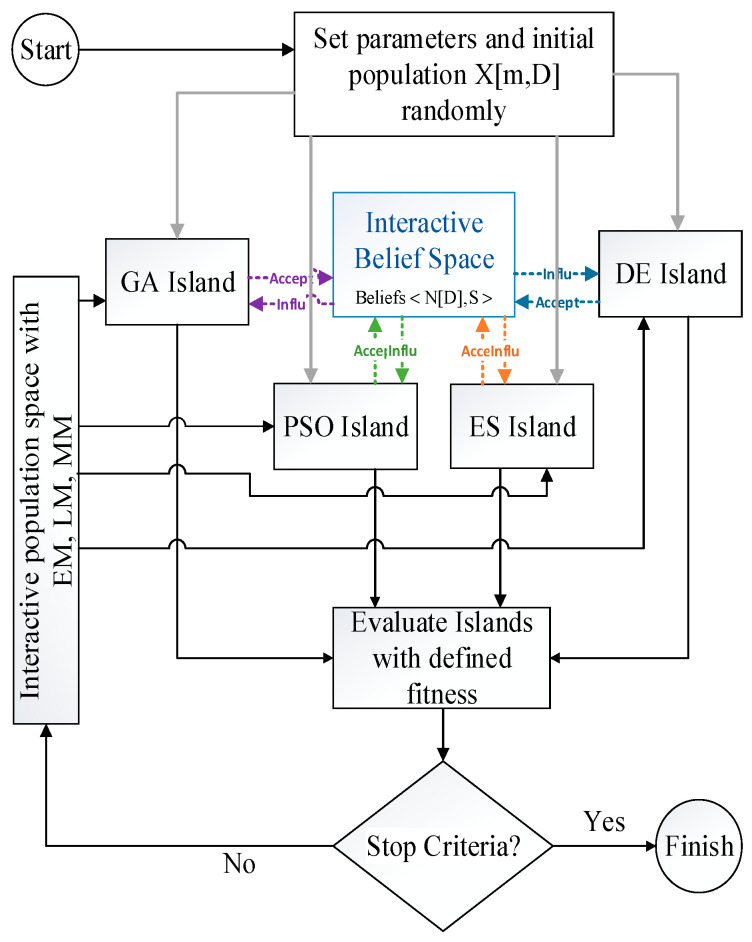
Flowchart of the MIC algorithm.

**Figure 2 sensors-22-06206-f002:**
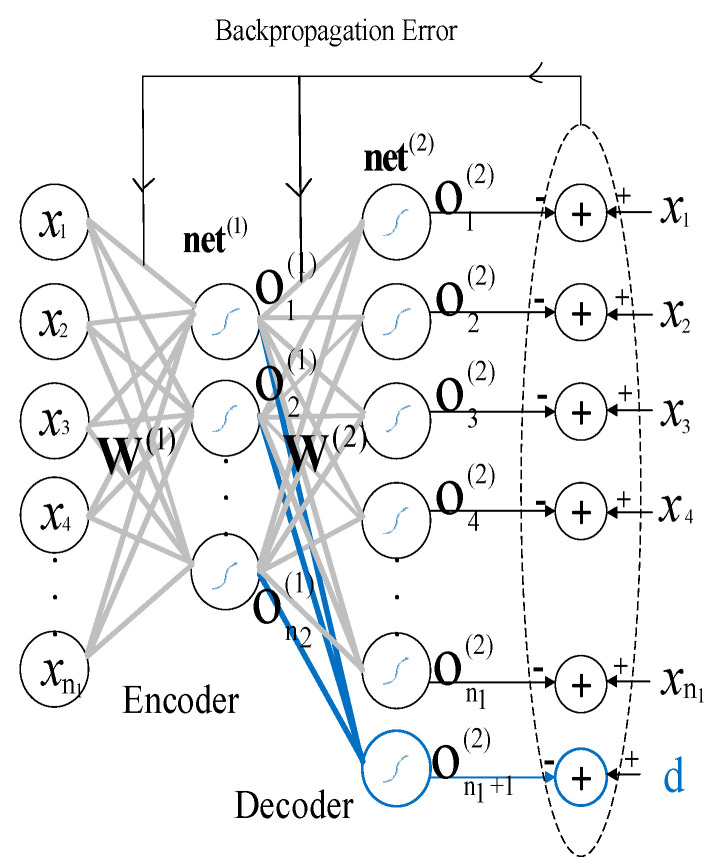
Schematic of the asymmetric auto-encoder [24].

**Figure 3 sensors-22-06206-f003:**
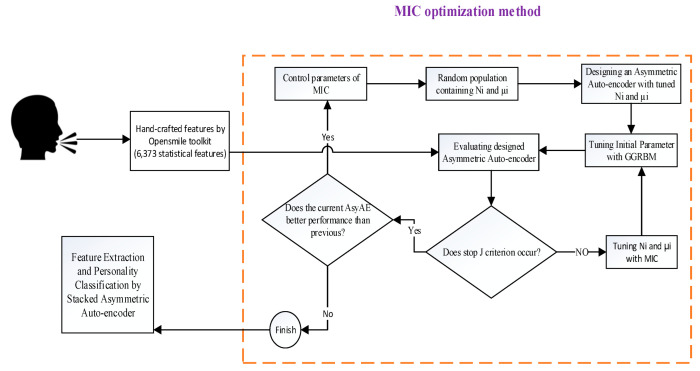
Flowchart of SA_AE_ hyper-parameter optimization.

**Figure 4 sensors-22-06206-f004:**
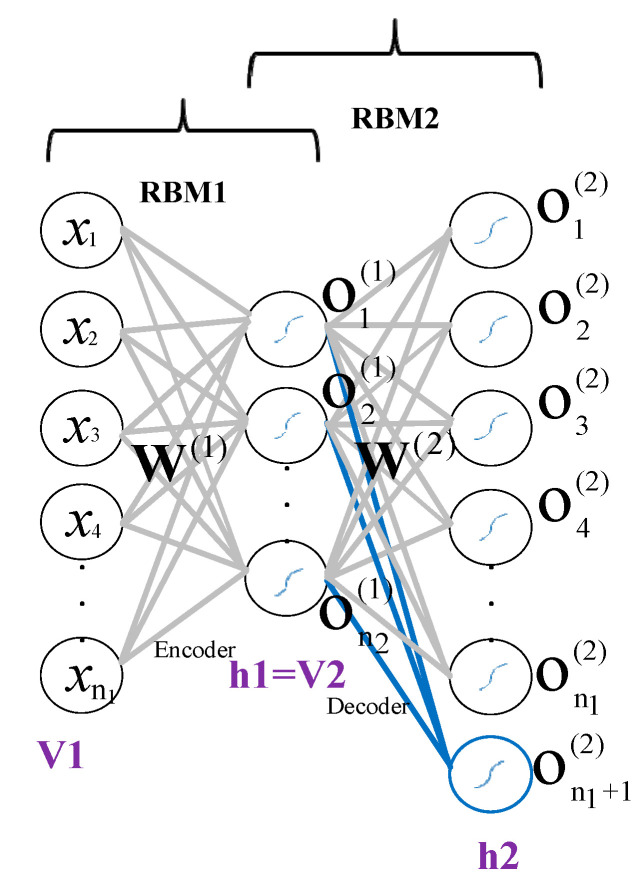
Converting auto-encoder to two RBMs for tuning the initial weights of the encoder and decoder layers.

**Figure 5 sensors-22-06206-f005:**
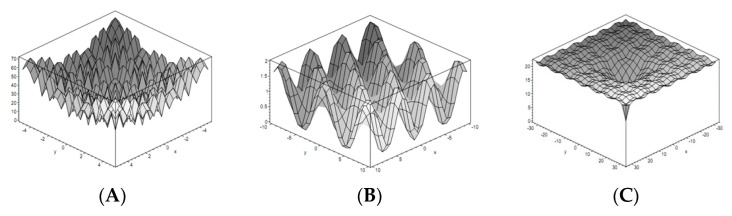
Benchmark functions (**A**) Rastrigin, (**B**) Ackley, and (**C**) Griewang.

**Table 1 sensors-22-06206-t001:** The 130 LLD features, including 65 LLD and 65 ΔLLD features [46].

4 Energy Related LLD	Group
Sum of Auditory Spectrum (Loudness)	Prosodic
Sum of RASTA-Style Filtered Auditory Spectrum	Prosodic
RMS Energy, Zero-Crossing Rate	Prosodic
**55 Spectral LLD**	**Group**
RASTA-Style Auditory Spectrum, Bands 1–26 (0–8 kHz)	Spectral
MFCC 1-14	Cepstral
Spectral Energy 250–650 Hz, 1 k–4 kHz	Spectral
Spectral Roll Off Point 0.25, 0.50, 0.75, 0.90	Spectral
Spectral Flux, Centroid, Entropy, Slope, Harmonicity	Spectral
Spectral Psychoacoustic Sharpness	Spectral
Spectral Variance, Skewness, Kurtosis	Spectral
**6 Voicing Related LLD**	**Group**
F0 (SHS & Viterbi Smoothing)	Prosodic
Probability of Voicing	Sound Quality
Log. HNR, Jitter (Local, Delta), Shimmer (Local)	Sound Quality
**Mean Values** Arithmetic Mean A^∆^, B, Arithmetic Mean of Positive Values ^A∆, B^, Root-Quadratic Mean, Flatness **Moments:** Standard Deviation, Skewness, Kurtosis **Temporal Centroid** A^∆^, B **Percentiles** Quartiles 1–3, Inter-Quartile Ranges 1–2, 2–3, 1–3, 1%—tile, 99%—tile, Range 1–99% **Extrema** Relative Position of Maximum and Minimum, Full Range (Maximum–Minimum) **Peaks and Valleys** ^A^ Mean of Peak Amplitudes, Difference of Mean of Peak Amplitudes to Arithmetic Mean, Peak to Peak Distances: Mean and Standard Deviation, Peak Range Relative to Arithmetic Mean, Range of Peak Amplitude Values, Range of Valley Amplitude Values Relative to Arithmetic Mean, Valley-Peak (Rising) Slopes: Mean and Standard Deviation, Peak-Valley (Falling) Slopes: Mean and Standard Deviation **Up-Level Times:** 25%, 50%, 75%, 90% **Rise and Curvature Time** Relative Time in which Signal is Rising, Relative Time in which Signal has Left Curvative **Segment Lengths** ^A^ Mean, Standard Deviation, Minimum, Maximum **Regression** A^∆^, B Linear Regression: Slope, Offset, Quadratic Error, Quadratic Regression: Coefficients a and b, Offset c, Quadratic Error **Linear Prediction** LP Analysis Gain (Amplitude Error), LP Coefficients 1–5 ^A^ Functionals applied only to energy related and spectral LLDs (group A) ^B^ Functionals applied only to voicing related LLDs (group B) ^∆^ Functionals applied only to ∆LLDs ^∆^ Functionals not applied only to ∆LLDs	

**Table 2 sensors-22-06206-t002:** Description of Three Benchmark Functions.

Name	Formula	Range	$\mathrm{Optimal}\;f\left(x\right)$
Rastrigin	$f\left(x\right)=10n+\overset{n}{\underset{i=1}{\sum }}x_{i}^{2}-10\cos\left(2{\pi x}_{i}\right)$	$-5.12<x_{i}<5.12$	0
Ackley	$f\left(x\right)=-20\exp\left(-0.02\sqrt{\frac{1}{n}\overset{n}{\underset{i=1}{\sum }}x_{i}^{2}}\right)-\exp\left(\frac{1}{n}\overset{n}{\underset{i=1}{\sum }}\cos\left(2{\pi x}_{i}\right)\right)+20+\exp\left(1\right)$	$-32\leq x_{i}\leq 32$	0
Griewang	$f(x)=\frac{1}{4000}\sum_{i=1}^{n}x_{i}^{2}-\prod_{i=1}^{n}\cos\left(\frac{x_{i}}{\sqrt{i}}\right)+1$	$-600<x_{i}<600$	0

**Table 3 sensors-22-06206-t003:** The Results of MIC Compared with Traditional GA, DE, PSO, and ES in Three Benchmark Functions (10D and 30D).

Benchmark Functions	Optimization Algorithm	AvI	AvP	SI	BOP	SD	SR (%)	AvI	AvP	SI	BOP	SD	SR (%)
		n=10						n=30					
Rastrigin	MIC by LM	83.4	6.7 × 10^−5^	61	7.2 × 10^−5^	4.7 × 10^−5^	100	307.1	7.8 × 10^−4^	254	4.8 × 10^−4^	2.4 × 10^−4^	100
	MIC by EM	120.5	7.1 × 10^−5^	101	8.8 × 10^−6^	5.8 × 10^−5^	100	321.5	7.4 × 10^−4^	187	1.8 × 10^−4^	5.6 × 10^−4^	100
	MIC by MM	572.5	9.9 × 10^−3^	131	4.9 × 10^−3^	4.3 × 10^−3^	100	2000	0.46	2000	6.1 × 10^−4^	1.32	60
	GA	617.1	1.2 × 10^−5^	324	6.8 × 10^−4^	4.6 × 10^−3^	40	1178.4	1.34	926	6.8 × 10^−4^	3.20	50
	DE	1000	0.22	1000	0.14	0.47	0	2000	4.72	2000	0.10	2.55	0
	ES	1000	3.48	1000	1.94	2.39	0	2000	37.4	2000	24.7	14.4	0
	PSO	985.7	0.82	857	6.8 × 10^−4^	0.71	20	2000	10.9	2000	3.13	5.60	0
Ackley	MIC by LM	467.2	4.4 × 10^−15^	355	4.4 × 10^−15^	0	100	1039.6	4.4 × 10^−15^	956	4.4 × 10^−15^	0	100
	MIC by EM	788.4	4.4 × 10^−15^	462	4.4 × 10^−15^	0	100	1154.8	3.1 × 10^−14^	937	4.4 × 10^−15^	1.9 × 10^−15^	100
	MIC by MM	725.2	1.4 × 10^−9^	324	3.5 × 10^−10^	9.2 × 10^−10^	100	2000	2.48	2000	2.24	0.20	0
	GA	957.5	7.3 × 10^−2^	565	7.2 × 10^−3^	1.58	70	1895.1	2.86	951	0.01	1.53	10
	DE	1000	1.69	1000	1.24	4.47	0	2000	5.35	2000	4.34	0.67	0
	ES	1000	4.96	1000	3.20	3.09	0	2000	5.43	2000	5.23	1.9 × 10^−1^	0
	PSO	557.5	8.6 × 10^−4^	344	6.2 × 10^−4^	4.1 × 10^−4^	100	839.5	4.5 × 10^−3^	162	8.8 × 10^−4^	7.9 × 10^−3^	100
Griewang	MIC by LM	154.7	6.2 × 10^−14^	38	1.2 × 10^−14^	2.8 × 10^−14^	100	106	8.4 × 10^−14^	92	8.7 × 10^−14^	4.4 × 10^−14^	100
	MIC by EM	171.2	8.4 × 10^−14^	43	6.4 × 10^−14^	1.5 × 10^−14^	100	489	1.1 × 10^−13^	94	9.1 × 10^−14^	2.5 × 10^−14^	100
	MIC by MM	775.4	3.3 × 10^−13^	146	9.6 × 10^−14^	3.2 × 10^−13^	100	2000	0.27	2000	9.1 × 10^−13^	0.16	20
	GA	909.8	0.09	84	0.8 × 10^−3^	0.18	10	2000	0.21	2000	0.09	1.1 × 10^−1^	0
	DE	337.4	9.1 × 10^−3^	44	7.3 × 10^−3^	1.1 × 10^−3^	100	993	0.01	588	7.9 × 10^−3^	8.8 × 10^−1^	70
	ES	1000	0.36	1000	1.2 × 10^−1^	0.20	0	2000	0.81	2000	0.76	0.19	0
	PSO	555.2	0.02	258	6.6 × 10^−2^	2.8 × 10^−2^	30	2000	0.77	2000	0.37	3.1 × 10^−2^	0

**Table 4 sensors-22-06206-t004:** Comparison with Other Published Methods in 30D. N/A means not available.

Methods	Benchmarks								
	Rastrigin			Ackley			Griewang		
	AvP	SD	SR%	AvP	SD	SR%	AvP	SD	SR%
Xin Zhao et al., 2022 [55]	2.1 × 10^−13^	4.1 × 10^−14^	100	8.2 × 10^−15^	1.3 × 10^−15^	100	3.78 × 10^−13^	1.7 × 10^−13^	100
Chentoufi et al., 2021 [49]	0.99	1.31	100	1.0 × 10^−15^	6.4 × 10^−16^	43	8.3 × 10^−4^	5.4 × 10^−4^	67
MIC_LM	7.8 × 10^−4^	2.4 × 10^−4^	100	4.4 × 10^−15^	0	100	8.4 × 10^−14^	4.4 × 10^−14^	100
MIC_EM	7.4 × 10^−4^	5.6 × 10^−4^	100	3.1 × 10^−14^	1.9 × 10^−15^	100	1.1 × 10^−13^	2.5 × 10^−14^	100
MIC_MM	0.46	1.32	60	2.48	0.20	0	0.27	0.16	20

**Table 5 sensors-22-06206-t005:** Comparison Results of Our Proposed Method with Other Works in the SPC Dataset in Terms of UA Recall % (Accuracy %).

Methods	Traits				
	Neu.	Ext.	Ope.	Agr.	Con.
Mohammadi et al., 2010 [43]	N/A (63)	N/A (76.3)	N/A (57.9)	N/A (63)	N/A (72)
Mohammadi et al., 2012 [63]	N/A (65.9)	N/A (73.5)	N/A (60.1)	N/A (63.1)	N/A (71.3)
Chastagnol et al., 2012 [64]	58 (N/A)	75.5 (N/A)	73.4 (N/A)	65 (N/A)	62.2 (N/A)
Mohammadi et al., 2015 [65]	N/A (66.1)	N/A (71.4)	N/A (58.6)	N/A (58.8)	N/A (72.5)
Solera-Urena et al., 2017 [66]	65.1 (64.7)	75 (75.1)	59.1 (58.2)	60.3 (60.2)	75.7 (75.6)
Carbonneau et al., 2017 [67]	70.8 (N/A)	75.2 (N/A)	56.3 (N/A)	64.9 (N/A)	63.8 (N/A)
Zhen-Tao Liu et al., 2020 [68]	N/A (69.2)	N/A (76.3)	N/A (74.7)	N/A (65.3)	N/A (73.3)
Our privuse work 2021 [24]	77.1 (76.9)	76.6 (72.9)	81.2 (70.4)	80.7 (68.7)	78.5 (69.5)
Proposed method	89.8 (80.5)	82.2 (83.4)	87.1 (84.7)	85.8 (76.2)	81.8 (72.6)

## Data Availability

Not applicable.

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
