# Peer review of "Hyper-Parameter Optimization of Stacked Asymmetric Auto-Encoders for Automatic Personality Traits Perception"

_sensors, 2022, doi:10.3390/s22166206_

Round 1

Reviewer 1 Report

This paper presents a hyperparameter optimization method for deep learning based on the cultural algorithm and parallel computing. The authors benchmarked the proposed method and compared its performance with the existing models on computation speed and escaping local optima. Experimental results showed that the new method outperformed others methods. Overall, the paper is well organized. The authors may consider using subsections to improve the structure and highlight the subtitles (e.g., change lines 84, 129, 182, 321 ... to subsection 2.1, 2.2, etc.). Also, minor writing issues exist throughout the paper (e.g., lines 31-32, verbs are in different tenses). Proofreading is recommended before resubmission.  

Author Response

Reviewer 1, Concern # 1: Overall, the paper is well organized. The authors may consider using subsections to improve the structure and highlight the subtitles (e.g., change lines 84, 129, 182, 321 ... to subsection 2.1, 2.2, etc.).

Author response: Thanks for your suggestion!

Author action: We have updated the manuscript by replacing subtitles with subsections.

Reviewer 1, Concern # 2: Also, minor writing issues exist throughout the paper (e.g., lines 31-32, verbs are in different tenses). Proofreading is recommended before resubmission.

Author response: Thanks for the careful review. In this regard, we went through the paper to remove typos and grammatical mistakes.

Author action: We proofed the manuscript. 

Reviewer 2 Report

This paper is a super parameter optimization of superimposed asymmetric automatic encoder based on automatic personality characteristics. This paper presents a new MIC method, which gets rid of the local optimal solution, accelerates the convergence speed, and significantly improves the accuracy of the calculation results. The new method proposed in this paper will be used to find the approximate optimal value of the super parameters of the asymmetric self encoder, which is our previous work in the aspect of automatic personality perception. However, there are still some minor problems to be improved.

1.In the third part of the data set and feature extraction methods, you can add more analysis and discussion

2.In the fourth part, combined with detailed data analysis, the MIC method is introduced in detail, and the advantages of this method are explained

3.In the fifth part, MIC is evaluated on two complex functions and compared with other traditional methods. The results show that MIC method is better, and it will be better to add more data to explain in this part

Author Response

Reviewer 2, Concern # 1: In the third part of the data set and feature extraction methods, you can add more analysis and discussion.

Author response: Thanks for the suggestion.

Author action: We updated the manuscript by adding more details in the section about the dataset and the feature extraction methods.

Reviewer 2, Concern # 2: In the fourth part, combined with detailed data analysis, the MIC method is introduced in detail, and the advantages of this method are explained.

Author response: We appreciate the reviewer for the given comment.

Reviewer 2, Concern # 3: In the fi(h part, MIC is evaluated on two complex functions and compared with other traditional methods. The results show that the MIC method is better, and it will be better to add more data to explain in this part.

Author response: Thank you for your suggestion. We added an extra benchmark function and its simulation results in this part to indicate the performance of our method better in comparison to other optimization methods. Also, we analyzed the results for n=10 and n=30 more accurately. Please see the "Simulations and results" section. 

Reviewer 3 Report

There are some contents, which need be revised.

 (1) The abstract is required to better describe the content of this paper.

 (2) Those related works and their relevance are required to analyze.

(3) Add the contents in the abstract of the paper and highlight the impact of the proposed work.

 (4)  The references are inadequate. I hope that the authors can add some new references in order to improve the reviews part and the connection with the literatures. For example, 10.3390/agriculture12060793; 10.1109/JSTARS.2021.3059451; 10.1007/s10489-022-03719-6; 10.1016/j.isatra.2021.07.017 and so on.

(5) Correct typological mistakes and mathematical errors. The paper is in need of revision in terms of eliminating grammatical errors, and improving clarity and readability.

(6) How about the computation complexity of the proposed method?

Author Response

Reviewer 3, Concern # 1: The abstract is required to be0er describe the content of this paper.

Author response: Thank you for your recommendation.

Author action: According to your comment, we have rewritten the abstract and added more details.

Reviewer 3, Concern # 2: Those related works and their relevance are required to analyze.

Author response: With appreciation for your suggestion, given that this article examines the HPO methods in order to find a proper one to optimize the hyper-parameters of stack asymmetric auto-encoder for automatic personality traits perception, the related works section is divided into two parts. The focus of the first part is on the recently published methods on neural network hyper-parameters tuning, regardless of the application in which it is used. Thus, the related works on the investigation of HPO in ML are summarized in the first part. Since the aim of our research is HPT of the SAAE to classify five personality traits from speech, the next part is related to studying HPO in machine learning methods applied in the field of personality trait perception. Considering that the change in the value of each hyper-parameter changes the values of the neural network parameters that affect the output of the network, and also examination of every possible combination of hyper-parameters is time-consuming, expensive and practically impossible, so studies have investigated the effect of adjusting/optimizing some of the most important hyper-parameters.

Author action: We updated the manuscript by adding the above descriptions and references to more recent articles.

Reviewer 3, Concern # 3: Add the contents in the abstract of the paper and highlight the impact of the proposed work.

Author response: Thanks for your careful review.

Author ac/on: We rewrote both the abstract and the conclusions to highlight the impact of our work.

Reviewer 3, Concern # 4: The references are inadequate. I hope that the authors can add some new references in order to improve the reviews part and the connection with the literatures. For example, 1) 10.3390/agriculture12060793; 2) 10.1109/JSTARS.2021.3059451; 3) 10.1007/s10489-022-03719-6; 4) 10.1016/j.isatra.2021.07.017 and so on.

Author response: We appreciate for these useful papers. We cited all of them in the manuscript. (Ref.[7], [12], [21], [34] ).

Author action: We have updated the manuscript by adding and replacing some references (new Refs: [2], [3], [7], [10], [12], [13], [14], [15], [16], [21], [32], [33], [34], [42], [55]). Reviewer 3, Concern # 5: Correct typological mistakes and mathematical errors. The paper is in need of revision in terms of elimina?ng grammatical errors, and improving clarity and readability.

Author response: Thanks for the careful review. In this regard, we went through the paper to check if there are any typos and grammatical mistakes.

Author action: Please check the corrections throughout the paper. Reviewer 3, Concern # 6: How about the computa?on complexity of the proposed method,

Author response: Great question! We answer this question by considering two aspects. 1. The number of deep neural network parameters is enormous, and calculating the value of these parameters is complicated work, not easily implemented, and requires computational systems with remarkable capabilities and large storage capacity. In this aspect, the deep neural network has inherent computational complexity. Therefore they are usually implemented on a GPU, and we used GPUs as well. 2. Our method is based on parallel computing and multi-island, where each island is an optimization algorithm. In other words, four basic optimization algorithms are implemented simultaneously with the same population in the Pirst iteration. Each island has its own evaluation process and generates its offspring. In the next iteration, proposed driving forces which discover the searching space smartly somehow use the advantages of all methods and ignore their disadvantages and increase diversity. So, the stop criterion occurs in lower iterations which mean speeds up convergence (please see Table 3 in the manuscript). In general, the use of deep neural networks has its computational complexity, and using GPUs helps speed up the calculations. On the other hand, the training of deep neural network parameters is done offline. This means that the calculation time is not taken into account during the network test. Although in the MIC method the calculations are a bit more than four basic islands, we paid attention to computational time in a reasonable time. The proof of this is the reduction in the number of iteration to reach the stop criterion. We hope our explanation has been complete and comprehensive enough to address your concerns. 

Round 2

Reviewer 3 Report

I have appreciated the deep revision of the contents and the present form of this manuscript. All my previous concerns have been accurately addressed. I think that this paper can be accepted.